# Metabolomics to Improve the Diagnostic Efficiency of Inborn Errors of Metabolism

**DOI:** 10.3390/ijms21041195

**Published:** 2020-02-11

**Authors:** Dylan Mordaunt, David Cox, Maria Fuller

**Affiliations:** 1Genetics and Molecular Pathology, SA Pathology at Women’s and Children’s Hospital, 72 King William Road, North Adelaide, SA 5006, Australia; dylan.mordaunt@sa.gov.au (D.M.); david.cox2@sa.gov.au (D.C.); 2School of Medicine, University of Adelaide, Adelaide, SA 5000, Australia

**Keywords:** inherited metabolic disease, enzyme, metabolite, mass spectrometry, diagnosis, screening, substrate, genetic variants, genetic disorder

## Abstract

Early diagnosis of inborn errors of metabolism (IEM)—a large group of congenital disorders—is critical, given that many respond well to targeted therapy. Newborn screening programs successfully capture a proportion of patients enabling early recognition and prompt initiation of therapy. For others, the heterogeneity in clinical presentation often confuses diagnosis with more common conditions. In the absence of family history and following clinical suspicion, the laboratory diagnosis typically begins with broad screening tests to circumscribe specialised metabolite and/or enzyme assays to identify the specific IEM. Confirmation of the biochemical diagnosis is usually achieved by identifying pathogenic genetic variants that will also enable cascade testing for family members. Unsurprisingly, this diagnostic trajectory is too often a protracted and lengthy process resulting in delays in diagnosis and, importantly, therapeutic intervention for these rare conditions is also postponed. Implementation of mass spectrometry technologies coupled with the expanding field of metabolomics is changing the landscape of diagnosing IEM as numerous metabolites, as well as enzymes, can now be measured collectively on a single mass spectrometry-based platform. As the biochemical consequences of impaired metabolism continue to be elucidated, the measurement of secondary metabolites common across groups of IEM will facilitate algorithms to further increase the efficiency of diagnosis.

## 1. Introduction

Inborn errors of metabolism (IEM) are a group of about 1000 relatively rare inherited disorders arising due to genetic variants that perturb the proper functioning of proteins requisite for cellular metabolism [1]. The result is a shortfall in energy production or pathological accumulation of macromolecules. These macromolecules would otherwise be metabolised, and their consequent accumulation causes the clinical manifestations associated with the specific disorder. A broad spectrum of symptoms and signs occur, with considerable variability in how a particular disorder presents even between individuals with the same IEM. Some conditions manifest characteristic features, and others present clinical features in which the index of suspicion is high, and yet others again remain entirely unsuspected and often come as a surprise to the treating physician when the laboratory reports the condition [2]. While metabolomics has been a powerful tool with a transformative effect for basic science [3], it also offers significant advantages (and potential pitfalls) in the clinical context [4]. The highly varied process and timeline of diagnoses of these disorders have been partly dependent on the rare nature of the conditions, the scarcity of information and the niche nature of laboratory diagnostics. Use of untargeted approaches early in the condition allows the clinician to make diagnoses earlier, and in many cases, reduce the overall cost of testing and provide the potential for functional validation of conditions diagnosed through other means, such as genome-based approaches [5]. Indeed, the relationship between molecular and biochemical testing and the role in screening, diagnosis and confirmation continue to evolve.

This article focuses on the improved diagnostic efficiency of IEM inherent in metabolomics, which comes from advances in technology scalability, streamlined operations management and the potential for greater ‘democratisation’ of the metabolomic screening approach.

## 2. Solving a Problem: The Changing Clinical Context of Metabolic Diagnostics

A clinician’s index of suspicion for a metabolic disorder is dependent on a variety of factors. There is the consideration of disease incidence with the oft-touted adage of “when one hears hoofbeats, think horses, not zebras” subverting efforts at early diagnoses of rare yet treatable disorders [2]. Other contributors, such as the environment and speciality of the practitioner, as well as proximity and access to diagnostic testing, considerably influence the ordering of diagnostic testing. Disease incidence differs by region and country, considerably varying the index of suspicion [2]. In some areas, certain conditions are so common that the general public may be aware of them, while in others, they are so rare that they are considered an academic curiosity. The rise of untargeted genomic testing in translational research environments has led to numerous examples of only one or two families with a particular condition being known to exist.

As an example, while a clinical geneticist, metabolic physician or paediatrician may consider mucopolysaccharidosis (MPS) in a coarse-appearing child with short stature and developmental delay, this diagnosis may never be considered by the midwife or general practitioner. A paediatrician may know to order urine glycosaminoglycans (GAG) in this case of iduronate-2-sulfatase deficiency causing Hunter syndrome. However, the differential diagnosis in an infant with these phenotypic features may be quite large and include many untreatable disorders. This example illustrates the complexity of IEM diagnosis and how it requires multiple tiers/iterations to identify convincing evidence of the underlying cause. In response, clinicians have become accustomed to ordering a ‘metabolic screen’ with multiple assays increasingly available to cover many conditions. This reflex of ordering a panel works well for a busy clinician trying to diagnostically solve an undifferentiated patient in a timely fashion and to ensure they do not miss the opportunity to offer early treatment.

Multi-disorder panels have become cemented in the workflow of clinicians in recent years, with multi-gene next-generation sequencing panels and genome-wide sequencing approaches. However, often the ‘test requestors’ are unfamiliar with functional validation of findings made on these panels, utilising biochemical investigations. Thus, multiplex metabolomics-based workflows solve the clinical problem of not being entirely sure of what is going on with the patient in front of them. It also assists with interpreting molecular findings from genomic testing, whilst being able to have confidence that a wide range of treatable conditions have been tested.

## 3. Traditional Tests for IEM

The choice of first-tier laboratory tests is somewhat limited and is dependent on the nature of the symptoms and signs (phenotype) being investigated (Figure 1). The potential range of symptoms for metabolic conditions are broad, and some of the characteristic features are listed in Table 1. Perhaps the single largest groups investigated for IEM are the asymptomatic newborn screening (NBS) population and the undifferentiated developmental delay (including autism) population. Outside these, typical presentations include the acutely unwell child with catastrophic symptoms such as refractory seizures, altered level of consciousness, muscle weakness, heart failure and liver and kidney failure [6].

### 3.1. Newborn Screening (NBS)

NBS involves screening neonates early in life to detect life-threatening or long-term health conditions to reduce morbidity and mortality. NBS began with phenylketonuria (PKU), once it was shown that early treatment spared intellectual disability, by a bacterial inhibition assay on dried filter paper blood spots (DBS) in the 1960s [7]. Principles of screening were described by Wilson and Junger in 1968 [8]. However, some conditions screened for in NBS programs do not meet these criteria, and thus an abbreviated set of principles have been proposed [9], which are:A treatment should be available that changes the outcome of patients based on early diagnosis.There should be a good understanding of the natural history of the condition.There should be a good understanding of which patients should be treatedA screening test should be available that is valid, reliable and acceptable to the public.

PKU is a condition which, when left untreated, results in intellectual disability, motor dysfunction and mental health issues [10]. Treatment with protein restriction and elemental supplementation has been shown to prevent these adverse outcomes. The same test used to diagnose the condition (amino acid levels in DBS) can also be used to monitor treatment outcomes. Since this time, other disorders have been added to NBS programs, including congenital hypothyroidism and cystic fibrosis. Expanded NBS by tandem mass spectrometry was introduced in the early 1990s [11]. This multiplied the scale of diagnosis from three to approximately 40 possible IEM, mainly amino acidopathies and fatty acid oxidation disorders, with a significant number of dietary congenital B12 deficiencies (via elevated C3-carnitine due to methylmalonic acidemia) in addition to the many genetic causes of methylmalonic acidemia and other detected organic acidurias [12,13].

Shifting NBS for PKU from a single-test method to a multiplexed mass-spectrometry method allowed for considerable expansion of the number of IEM that could be screened for at birth. Many of these had clear benefits from screening, as early diagnosis before the onset of symptoms enabled therapeutic intervention, improving health outcomes for infants [10]. However, some of these conditions are opportunistically screened, and perhaps would not be possible to justify a specific screening program based on health economic impact [14].

NBS has advanced substantially over the past ten years. Numerous applications of metabolomic-based NBS have been piloted, introduced or proposed in recent years [15,16]. The US Health Resources and Services Administration Recommended Uniform Screening Panel has currently mandated 57 conditions amenable to mass screening of neonates, including some lysosomal, peroxisomal as well as non-metabolic genetic disorders, such as spinal muscular atrophy and severe combined immunodeficiency [13]. However, issues have arisen, such as the benefit of screening and ‘medicalisation’ of variant-normal physiology. For example, 3-methylcrotonyl-CoA carboxylase deficiency causing methylcrotonylglycinuria is detected by 3-hydroxyisovalerylcarnitine in DBS and surprisingly found to be the most common organic aciduria detected by expanded NBS [17]. Nonetheless, there remained debate for some time as to whether this was a disease-associated biochemical phenotype and in recent years, it has been found to have a highly variable age of symptom onset, thus raising debate of its suitability for NBS. While many conditions can be easily added, this example demonstrates the need for a rigorous process of vetting candidates before approval.

New York State’s introduction of NBS for Krabbe disease (globoid leukodystrophy, due to β-galactosylceramidase deficiency) has followed a similar course with higher numbers of individuals being detected than are clinically affected [18]. Of those that are identified, the benefits of early treatment with haematogenous stem cell transplant remain questionable. Other issues have also surfaced with the addition of additional IEM to NBS programs by tandem mass spectrometry (MS/MS), including false positives and the identification of late-onset phenotypes in the asymptomatic neonate [19,20,21].

### 3.2. Acute Metabolic Presentation

A child presents with paroxysmal events, such as seizures or abnormal movements. Initial laboratory assessment may show reduced blood sugar, abnormalities on blood gas, such as metabolic acidosis with an increased anion gap due to lactic acidosis. Non-specific markers may also be abnormal, such as liver enzymes or other features, such as coagulopathy, being present. Somewhat more specific testing such as ketones or ammonium may be elevated or ‘inappropriately normal’ (e.g., hypoketotic hypoglycaemia), which allows for directed attention to specific metabolites in further testing (since many potential genetic causes underlie these abnormalities).

In this example of an acute metabolic acidosis, the approach could consider undertaking metabolic screening with plasma amino acids and urine organic acids. Specific metabolites could also be requested by a metabolic physician, such as urine orotate or succinylacetone. These could be arranged in tandem, however in most situations, these reflex pending results from the initial ‘screen’. Some centres have moved away from employing untargeted gas chromatography (GC)-based urine organic acid analysis towards semi-targeted liquid chromatography–electrospray ionisation (LC-ESI)-MS/MS analysis. This approach, first published by Pitt et al. [22], has been demonstrated to have similar sensitivity and specificity to GC-MS for conditions resulting in elevated organic acids while enabling multiplexing for additional conditions, thereby increasing the utility for diagnosis. Saving examples with specifically elevated metabolites in plasma amino acids or urine, most patients presenting like this will not initially have a metabolic diagnosis identified. In parallel, chromosomal analysis through karyotype or microarray, or increasingly multi-gene or genomic sequencing, will occur. The future will likely witness a rapid turnaround of both genomic and metabolomic data for diagnosis, functional validation and subsequent measurement of treatment response.

### 3.3. First-Tier Tests

Clinical phenotypes assist in selecting the appropriate laboratory investigations when ruling out IEM. Determining whether the condition is likely due to defects in small molecule metabolism (disorders of amino acids, organic acids, purines and pyrimidines, the urea cycle, mitochondrial energy metabolism) or defects of organelle metabolism (lysosomal or peroxisomes) will assist in the diagnostic pathway [23], as current approaches are targeted and semi-targeted (Figure 1, Figure 2 and Figure 3).

In the case of acutely presenting conditions, the approach to screening begins with plasma amino acids, plasma acylcarnitine profile and urine metabolic screen (amino acids, organic acid, GAG, oligosaccharides). Turnaround time for these tests varies from a day to weeks, significantly influencing management and parallel investigations. Some conditions present with a characteristic pattern on these initial tests, for instance, raised plasma alloisoleucine and branched-chain oxoacids in the urine of maple-syrup urine disease. Other conditions present with less specific patterns, where experienced and well-established laboratories may detect the presence of an atypically elevated metabolite. This may not be obvious to others. Examples include the presence of elevated 3-methylglutaconic acid in Barth syndrome or metabolites related to tyrosinemia type I in a urine organic acid screen. Urine organic acid analyses have utilised GC-MS-based approaches in an untargeted fashion with post hoc filtering of well-known metabolites [6]. These techniques have often been the core of a so-called metabolic screen, in addition to amino acids, total GAG and creatine metabolites. Hundreds of metabolites are routinely detected in the urine metabolic screen, particularly amino acids, organic acids and metabolites related to the citric acid cycle, amongst others. Urine organic acids is not an equally sensitive method across labs, which may result in missed diagnoses. A problem evident in metabolomics and well known from urine organic acid testing, is the problem of metabolites of unknown origin and variability of assays between laboratories.

Lysosomal enzymology performed on leukocytes is a first-tier line of investigation for phenotypes such as cardiomyopathy and young-onset dementia [2]. This remains the case in childhood and adolescence; however, with a large number of possible causes of cardiomyopathy, testing in this group has shifted towards multi-gene NGS panels [24]. A similar but distinct group of conditions involve very-long-chain fatty acids (VLCFA) analysis, a group of metabolites utilised to investigate suspected peroxisomal disorders such as rhizomelic chondrodysplasia punctata (which presents with typical skeletal features) or more commonly, as a screening test in the context of a spectrum of symptoms, such as young-onset dementia. In most centres, this is the initial diagnostic test; however, centres with the ability to detect pipecolic acid in the urine may reflex to VLCFA [25]. The current GC-MS method for VLCFA analysis tends to be laborious, time-consuming and expensive to perform [6]. Peroxisomal disorders is a specific example discussed in a statement by the American College of Medical Genetics and Genomics (ACMG), where the complexity of differentiating specific sub-types lends itself well to early molecular testing, particularly relevant with the introduction of x-linked adrenoleukodystrophy to the recommended uniform screening panel in the United States [26].

There are additional specific first-tier tests which are requested less frequently due to regional laboratory availability. Disorders of creatine metabolism present predominantly with developmental and neurological disorders and are often treatable [6]. Disorders of nucleoside metabolism vary widely in their presentation from multiple congenital abnormalities through to significant neurodevelopmental disorders and are relatively undifferentiated, making them a useful target for a screening approach [6]. Polyol metabolism disorders include transaldolase deficiency, a condition presenting with liver and clotting abnormalities [27,28].

The final significant group of conditions involve protein glycosylation. Transferrin isoform analysis can be performed by numerous methods and is used to screen for congenital disorders of *N*-glycosylation [6,29,30]. Use of this as a first-tier test varies widely, dependent on access to the test and local practice. Despite the relative rarity of many of these disorders, the reason for inclusion is the treatability of some of these conditions. It is noteworthy that the transferrin isoform assays have often been established for monitoring relapse of alcoholism and thus may not be designed, analysed, interpreted or reported with consideration for congenital disorders of *N*-glycosylation [31]. Apolipoprotein-CIII isoform analysis is used to screen for congenital disorders of O-glycosylation [32,33,34]. These are a much less well-described group of conditions and often have distinctive skeletal/connective tissue phenotypes in which clinical phenotyping with molecular testing are more likely to be the pathway to diagnosis, e.g., multiple exostoses present with this characteristic pattern of skeletal anomalies. Mass spectrometry is now typically used for the measurement of the isoforms [34].

### 3.4. Second-Tier Testing with Enzymology and Metabolites

Second-tier testing typically follows a positive first-tier test and is nominally based on refining the exact IEM. For the majority of IEM arising due to an enzyme deficiency, the standard is to measure the enzyme activity in cells, blood or biopsy tissue. The gold standard of enzyme confirmation from a biopsy is often impractical or unethical when reliable biomarkers or molecular testing offer an adequate alternative. For many metabolic conditions, specific enzyme assays exist to validate either molecular- or metabolite-based diagnoses. Examples include the diagnosis of Lesch–Nyhan syndrome (hypoxanthine-guanine phosphoribosyltransferase) reflexed from urine purine and pyrimidines, congenital disorder of glycosylation-1a (phosphomannomutase) from transferrin isoforms and neuraminidase and β-galactosidase activity to differentiate galactosialidosis and sialidosis following an abnormal oligosaccharide pattern. Multiplex enzymology performed by mass spectrometry and digital microfluidics has been described, both of which are being used for NBS programs [35,36]. Although the mass spectrometry-based enzymology has the benefit of scalability and the ability to easily add additional disorders, the digital microfluidics presents an attractive, cost-effective option [36]. Metabolite analysis can also be considered, such as succinylacetone in tyrosinemia type 1 [37] and glucosylsphingosine in Gaucher disease [38]. Another typical example would be differentiating sub-types of hyperphenylalaninemia, where pterin profiling would be undertaken for differentiating causes related to pterin metabolism from phenylalanine hydroxylase since treatment is different. Some of these are reflex tests, for instance, plasmalogen profiling in the context of abnormal VLCFA.

Challenges remain in the goal of developing whole metabolome screening methods. For instance, disorders with no specific biomarker, such as Niemann–Pick type C disease. Plasma oxysterols represent useful screening metabolites for this group of conditions [39,40], but they are non-specific. Whereas lysosphingomyelin-509 is more specific and therefore more useful diagnostically, it is also elevated in acid sphingomyelinase deficiency, although levels are much lower [41].

## 4. Molecular Testing

### 4.1. Confirmatory Diagnostic Testing

Molecular and cytogenetic testing has a complicated relationship with IEM. The earliest diagnostic genetic methods for diagnosing genetic conditions were biochemical, such as for alkaptonuria and PKU [42,43]. Biochemical markers were used for early linkage studies [44], and some of the earliest genes positionally cloned were indeed metabolic genes [45]. Genomic sequencing technologies have significantly scaled the range and number of disorders diagnosable in the genetic laboratory, however not without considerable challenges, particularly the interpretation of variants of uncertain significance (VUS) [46].

Molecular testing strategies depend somewhat on environment and funding arrangements, often taking a tiered approach and are dependent on the conditions tested. In the context of a known biochemical diagnosis with one candidate gene, a single gene is usually interrogated, e.g., Fabry disease, a small gene of seven exons. In some conditions, there are common variants, either causing the condition generally or as a founder variant in a specific geographical region, ethnicity or family. In this context, genotyping technologies are still useful, such as Sanger sequencing, allele-specific amplification and restriction enzyme digestion. A variety of high-throughput genotyping technologies for conditions with common variants also exist. To this end, MassArray is a technology employing a mass spectrometry (MALDI-TOF) platform coupled with termination primer amplification and bioinformatic techniques [47]. An example of a testing strategy utilising high-throughput genotyping is genotyping for common *CFTR* variants following positive immunoreactive trypsinogen for cystic fibrosis from NBS, which can detect up to 90 variants within the gene [48].

In conditions for which the causative gene from the biochemical phenotype is poorly defined or in metabolic disorders that are not able to be biochemically sub-typed (e.g., peroxisomal biogenesis disorders), an NGS panel undertaking post hoc filtering for genes of interest (a ‘panel’) can be informative. An example is maple-syrup urine disease, which has multiple genes linked with an indistinguishable metabolite profile [49]. In practice, the choice of the underlying assay is often linked with workflow, since post hoc filtering allows for the creation of a virtual gene panel. A larger group of conditions that have non-specific clinical and biochemical phenotypes, such as many of the mitochondrial disorders, untargeted and semi-targeted NGS, are currently the de facto second-tier investigation [50]. In the absence of family history, molecular testing is usually confirmatory. Increasingly, conditions emerge for which molecular testing has been the initial step, followed by orthogonal validation with biochemical methods. In this sense, molecular testing has not supplanted biochemical investigations; instead, the role has changed.

### 4.2. Biochemical Validation of Molecular Diagnoses

Genomic diagnoses create the inherent need for biochemical validation. One of the earliest examples followed the introduction of chromosomal microarray, which allowed the identification of many patients with X-linked steroid sulfatase deficiency. The diagnosis in these individuals could be corroborated by measuring enzyme activity and determining that this was below normal levels [51]. More recently, genome-wide sequencing studies and clinical diagnostics have highlighted unsuspected metabolic diagnoses in which metabolite or enzyme diagnostics serve to provide orthogonal validation [52].

### 4.3. The Era of “Exome/Genome-First”

Increasingly, acute, complex and undiagnosed diseases are referred for untargeted genomic testing rather than iterative molecular and biochemical testing (Figure 2). Thus the paradigm presented in Figure 1 is no longer the only approach. As the “exome-first” approach has evolved, two main challenges have become apparent—there are many VUS in metabolic pathway genes, in which there is limited ability to ascertain pathogenicity at present. A lack of knowledge and training around biochemical methods leads to a lack of consideration of IEM VUS in the phenotype. Another phenomenon has also been associated with both the introduction of genomic and metabolomic approaches, which is the increasing recognition of non-classical, mild and attenuated expressions of disease. As the economics of both genomic and metabolomic diagnostic approaches improve, these methods are likely to offer a synergising effect when ordered in tandem (Figure 3).

## 5. Improving the Efficiency of Diagnosis with Metabolomics

Evidence is pointing towards three principal utilities in clinical practice for metabolomic methods. Firstly, orthogonal (‘functional’) validation of molecular discoveries, mainly VUS. Secondly, profiles can be utilised in a multivariate model for diagnostic classification. Thirdly is screening for IEM, either in isolation or in tandem with molecular methods [52]. NGS heightened the well-recognised issue of functional validation of VUS, and as the number of new conditions, genes and variants discovered grows, so does the demand for methods that can provide functional validation. Fortunately, in the metabolic domain, metabolites and enzyme activity are a relatively accessible form of functional analysis not dissimilar to flow cytometry in the immunology field.

With all of these technologies, it is possible to glean additional information and thus move away from a single analyte per diagnosis, to a ‘profile’ or classification model approach [53,54,55,56]. This has existed to a limited extent for urine organic acid analysis and peroxisomal disorder diagnostics. However, the number of analytes has not been enough to warrant non-human classification methods. As the number of analytes scales to hundreds or thousands, the requirement for automated methods of supervised classification provides an opportunity to gather additional information, with the whole being more significant than the sum of the individual parts. These circumstances have been long-established in NBS with incremental improvements to existing methods [15,16,18,19,20,57]. As has been described, by switching the diagnostic method for the single disorder, PKU, over to mass spectrometry, dozens of other of conditions have been able to be simultaneously tested in the NBS, enabling early- and pre-symptomatic detection of these conditions, affording better health outcomes.

Untargeted methods are being employed to increase the number of conditions screened for in the clinic, based on DBS, urine, plasma and CSF samples [52,58,59]. This collapses many of the second-tier biochemical tests into a single test, with molecular testing often occurring in parallel and increasing the total number of conditions diagnosable. Screening for mucopolysaccharidoses with mass spectrometry is an illustrative example. Disorders of GAG metabolism present with a variety of symptoms including developmental delay, musculoskeletal issues and in some cases, dementia-like features. Eleven known enzyme deficiencies give rise to individual types and sub-types of MPS. Until recently, the heuristic for MPS diagnosis depended somewhat on the index of suspicion, due to the simultaneous importance of making an early diagnosis but the relative non-specificity of early features to non-specialist test requesters. These clinicians would typically order a ‘metabolic screen’, which in many laboratories involved the tests mentioned above, including urine total GAG, usually measured with a quick spectrophotometric assay, employing dimethylene blue [60]. This method’s major strength remains its simplicity and low cost. However, limitations include it being a screening test with limited precision, poor sensitivity and a significant false positive and false negative rate. If total GAG is raised, the next step is GAG electrophoresis with abnormal patterns suggesting groups of MPS but not individual types or subtypes. More recently, LC-ESI-MS/MS has been introduced replacing traditional urinary GAG analysis [61,62,63], and one method that enables screening for a specific MPS sub-type by first-line testing [61].

### 5.1. Diagnostics for Ultra-Rare Disorders

Many classic metabolic disorders are thought to be ultra-rare, with examples such as Farber disease being reported in about 100 patients worldwide. Although enzymology is available for this condition, this would usually be performed in the context of family history, a confirmatory context or in extremely uncommon cases where the condition is suspected. Attenuated cases are yet to be described, possibly linked with the lack of clinical specificity of the presentation. Metabolite diagnosis is now possible while being multiplexed with other sphingolipids [62], which is a useful screening test but not specific for the condition, with a number of conditions associated with elevations, particularly Niemann–Pick Type C disease [62]. Lack of specificity of the biochemical marker is not critical with the availability of orthogonal molecular testing. Thus, a condition with a reported frequency of less than 1 in 10 million could be routinely screened for as part of an expanded sphingolipid screen. This is another example of the power of metabolomics to test for conditions that would otherwise be too rare to warrant screening for in the single IEM paradigm.

Indeed, the problem that some conditions are too rare to sustain highly specialised biochemical tests in a clinical diagnostic laboratory has led to many tests only being able to be offered in a research environment. Multiplexing of diagnostic metabolites and enzymology using metabolomics, will hopefully reverse this trend and re-enable access to diagnostic biochemical testing for these ultra-rare conditions and contexts.

### 5.2. Mass Spectrometry, Cheminformatics and Machine Learning

Mass spectrometry is being increasingly utilised for metabolomics, beginning with screening for a select group of organic/amino acidemias and fatty acid oxidation disorders in the late 1990s, followed by the introduction of GAG, oxysterols and oligosaccharides [63,64]. Adoption of LC-ESI-MS/MS has begun to accelerate in many laboratories, with improvements in diagnostic efficiencies as well as reductions in the cost and implementation of instrumentation. Many existing assays combine multiple analytes and analyte ratios to create variables which assist in the diagnosis of a condition. The scientist/clinician then interprets these analytes, and a conclusion is drawn from prior knowledge of the domain. As the number of analytes scales, both the range, number and complexity of diagnoses become humanly more challenging to manage. High-resolution technologies are also on the uptake, which comes with a new set of obstacles and opportunities, allowing for thousands of putative analytes to be measured in a semi-quantitative state.

The high-dimensional nature of the data being dealt with requires scalable pre-processing platforms, sophisticated multivariate analysis methods and a shift from the ‘one analyte per diagnosis’ approach of old to the metabolome-wide profiling of new [65]. These technologies also allow exploration of novel methods and translation to less expensive, accessible or high-throughput instrumentation. An example is the translation of urinary oligosaccharide testing from TLC to MALDI-TOF to LC-ESI-MS/MS [66,67,68,69,70,71]. Several open-source and commercial solutions have been developed for these problems, with notable examples of the former in the metabolic domain being XCMS and Metaboanalyst [72,73]. Dimension reduction techniques such as t-distributed stochastic neighbour embedding (t-SNE), combined with machine learning methods such as the support vector machine (SVM) algorithm, allow for the development of classification models with small numbers of affected cases, that provide rapid decision support for the interpretation of metabolomic profiles [74,75,76]. These same algorithms (e.g., SVM) can also be used to select for a small number of the highest performing features, amongst thousands of putative metabolites, for transfer to lower resolution assays.

With more substantial amounts of data being produced by these multiplex assays, machine learning tools facilitate reproducible and understandable models of prediction (classification and regression) [77]. These techniques take an entire metabolic snapshot of the metabolome and range from several to hundreds of analytes, to classify the sample in order to arrive at a diagnosis [54,56]. These cheminformatic methods are sufficiently reproducible for screening purposes, a requirement for quality control that is difficult to achieve with manual human analysis alone [54].

### 5.3. Biomarker Discovery

A biomarker is defined as an objective measure of the presence or severity of a non-physiological state, which can be measured accurately and reproducibly [78]. For many years, semi-targeted approaches on GC-MS have been exploited [79,80,81], which has been filtered for primary, quantitative diagnostic data. Much of the data produced in urine organic acid analyses are not routinely considered in terms of biomarker discovery, and this information remains unmined. In one example, additional information on urine organic acid analysis was derived that could inform on treatment compliance [82]. Data generated from semi-targeted LC-MS/MS platforms could also be interrogated [54,56], were this to be linked with data on new diagnoses made through other methods, such as genomics [83]. The power here rests with creating linked datasets and bi-directionally mining them, as a form of systematic examination. As with the advent of untargeted genomic testing (whole exome and genome sequencing), untargeted and semi-targeted metabolomic approaches provide a platform for the development of new disease biomarkers in situ [84], that is development, validation and repurposing for new clinical purposes while continuing to measure them for their existing intentions [83]. This underpins the notion for rapid translation from research into practice.

While many of the methods in this space have been developed for diagnosis and monitoring of constitutional errors of metabolism, the same techniques lend themselves to the measurement of perturbations in acquired (somatic) errors of metabolism seen in disease states such as cancer and in differentiating features of tumours. Indeed, some of these conditions even involve the same genes, an example being that biallelic mutations in *IDH1* cause hydroxyglutaric aciduria, whereas mono-allelic hypermorphic mutations cause a druggable form of brainstem glioma [85,86]. Recent examples of diagnostic improvements from the mining of metabolomic data in known conditions include the identification of profiles specific to urine samples from patients who are documented to be adherent and non-adherent to therapies in PKU [82]. Patients’ non-adherent to treatment had elevated phenylalanine metabolites with lower urinary excretion of compounds associated with the metabolism of tyrosine by microbiota activity, as well as several metabolites related to inadequate nutrient intake. An unknown metabolite was strongly associated with phenylalanine excretion, which was subsequently identified as imidazole lactic acid. Ultimately, this study showed the potential of urine profiling for non-invasive monitoring of individual PKU patients.

In practice, the subsequent reuse of metabolomic data collected for primary diagnostic purposes has already yielded benefits. Urine, plasma and CSF metabolomic assays have been employed, revealing significant findings on clinically analysed samples [58,85,86,87,88]. These insights include new diagnostic biomarkers [87,88,89], a broader range of phenotype in adenylosuccinate lyase deficiency [90], insights around pre-analytical factors [91], primary metabolic variations from drug-related changes [92,93,94] and a better understanding of the underlying pathobiological mechanism of disease [95,96,97].

## 6. The Future

Limitations of current technologies mean that metabolomic testing is not truly pan-system, but it is progressing, and many laboratories are cognisant of the approach. Undoubtedly, conditions and diagnostic metabolites/profiles are yet to be determined, and on the contrary, others are well-defined. Conditions such as PKU still have incremental value, added through expanded metabolite screens, and new disorders continue to be discovered. Bioinformatics workflows remain non-standard and often operate with only quasi-reference to either analysis of other samples but also other diagnostic methods, such as nucleic acids and proteins. The strength of the metabolomic approach is not only discriminating a condition but also reflecting on the primary-, second- and third-order effects of a metabolic perturbation, and the resulting response to treatment. The future of both metabolomic- and genomic-based diagnoses involves integrating these technologies to make and validate diagnoses, to prognosticate as well as monitor disease progression and response to therapy.

## Figures and Tables

**Figure 1 ijms-21-01195-f001:**
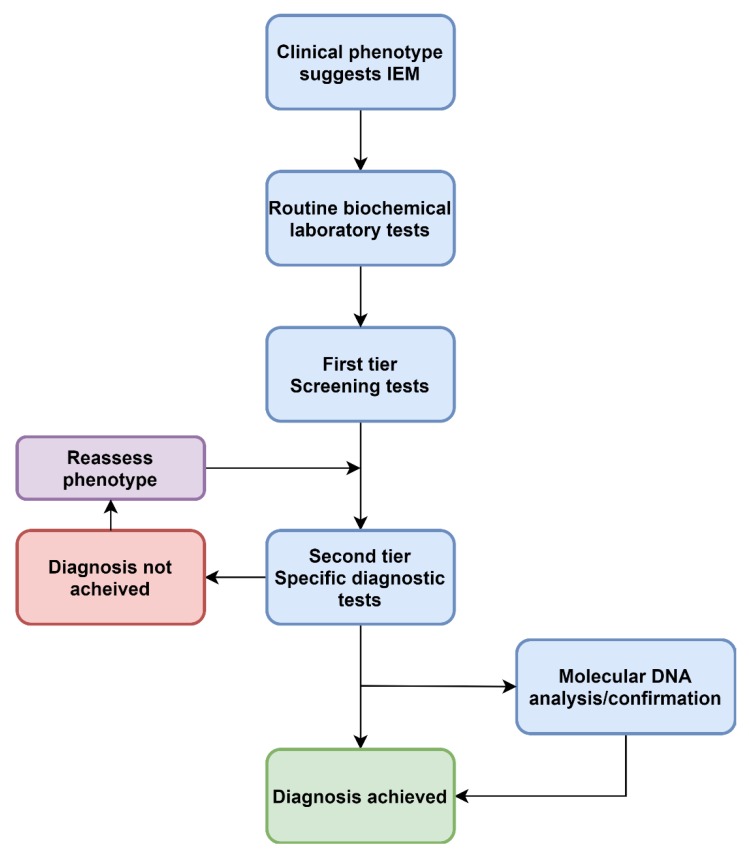
A traditional approach to diagnosing IEM. This describes a standard approach to screening that has not changed in decades.

**Figure 2 ijms-21-01195-f002:**
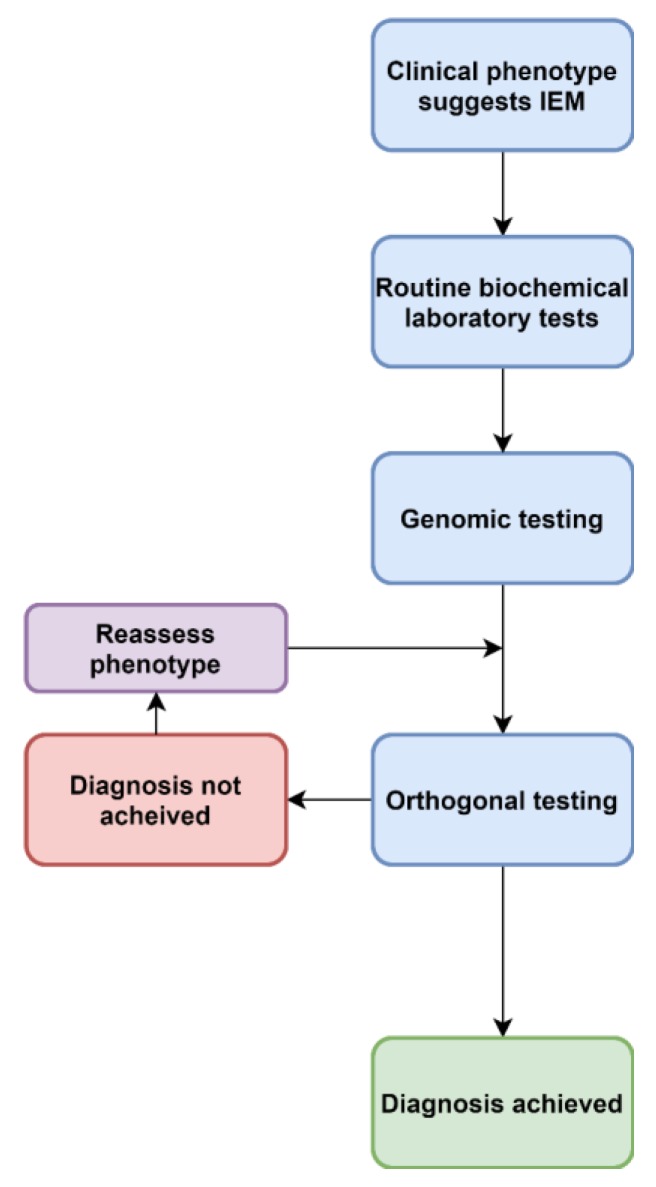
An evolving approach to diagnoses of IEM.

**Figure 3 ijms-21-01195-f003:**
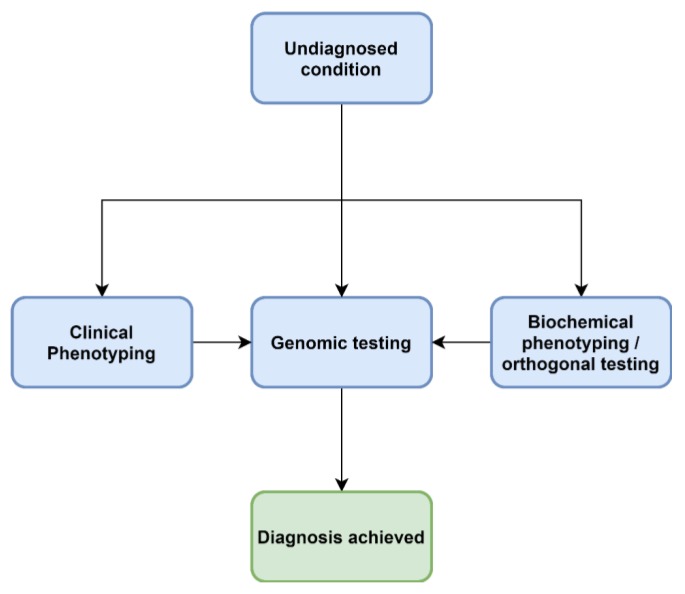
A future approach to diagnoses of IEM.

**Table 1 ijms-21-01195-t001:** Main clinical manifestations of inborn errors of metabolism (IEM).

System	Symptom/Sign
Neurological	Ataxia, seizures, arthrogryposis, movement disorders, myopathy, hypotonia, encephalopathy, myelopathy, leukodystrophy, intellectual impairment, autism, regression (loss of acquired skills), white matter changes, hydrocephalus, acroparaesthesia
Gastrointestinal	Hepatomegaly, liver failure and fibrosis, jaundice, hyperammonemia, splenomegaly, abdominal pain, gastrointestinal insufficiency, nausea/vomiting
Cardiac	Arrhythmias, valve disease, cardiomyopathy
Endocrine	Adrenal failure, hypoglycaemia, hypocalcaemia, ambiguous genitalia, hypo/hyperketosis
Skeletal	Joint restriction, dysostosis multiplex, short/tall stature, osteochondrosis
General	Failure to thrive, dysmorphism, hypo/hyperhidrosis
Eye	Cataracts, retinopathy, gaze palsy, lens dislocation

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
