# Peer review of "Metabolomics to Improve the Diagnostic Efficiency of Inborn Errors of Metabolism"

_ijms, 2020, doi:10.3390/ijms21041195_

Round 1
Reviewer 1 Report
Mordaunt et al have provided a well-written and comprehensive review of metabolic screening and testing related to inborn errors of metabolism from the perspective of suspected IEM in a patient. The content is appropriate and considers new biochemical testing and it's possible role in this process (untargeted metabolomics); however, the missing piece is discussion of molecular testing earlier in this diagnostic process (discussed in more detail below).
Considering current healthcare, one aspect that is not addressed is the decreasing number of clinically available tests. The need for tests to be clinically available and not research only is critical for patients; however, because of the rarity of many conditions, these tests are often research only and many tests that were once available clinically are no longer --because labs cannot afford to keep the tests active in their test menus. As testing becomes focused on molecular findings first, this will be an added concern because of inability to determine if variants are pathogenic.
Other points that are not really mentioned:
1. Lack of specificity of a specific biomarker to one disease is not critical if molecular testing is done in parallel (or as reflex).
2. Newborn screening has improved substantially in the last 10 years. The challenge is that the standard testing that is done when NBS is normal (UOA, PAA, ACP) do not provide much diagnostic coverage beyond the current NBS, which covers ~50 conditions. Of course, the NBS varies across countries.
3. As exome first has become the approach to testing--- two outomes/challenges have surfaced:
a. Many VUS are identified in metabolic pathway genes with inability to determine pathogenicity
b. Often, lack of knowledge, training, and experience in metabolism leads to lack of consideration of VUS in the phenotype
4. Metabolomics and exome have identified patients with milder metabolic disease who otherwise would have been missed because the phenotype was not classic. (GABA-transaminase deficiency, for example, as shown in Kennedy et al 2019, among others) Thus, the phenotype for metabolic disease is more broad and if limited to what we think we know, we will miss the diagnosis.
A few other comments:
1. UOA methods are not equally sensitive across labs, which may result in missing a diagnosis
2. Line 192: update/add references with ACMG lab guidelines for peroxisomal disorders
3. Reference 61 related to Farber-- C26 not so specific to Farber /acid sphingomyelinase deficiency. A reasonable screen but non-specific. Screening for non-specific molecules is still informative—for example, indicating peroxisomal dysfunction-- but not necessarily a specific gene. Narrowing the focus is quite helpful toward diagnosis.
5. Figure 1 describes a very standard approach to screening that hasn't changed in decades, without consideration of molecular testing/screening as part of this approach (exome), in addition to impacts of current NBS. Exome first is commonly used.
6. X-ALD is now being added to NBS in the U.S.
Author Response
Response to Reviewers’
Reviewer 1
Mordaunt et al have provided a well-written and comprehensive review of metabolic screening and testing related to inborn errors of metabolism from the perspective of suspected IEM in a patient. The content is appropriate and considers new biochemical testing and it's possible role in this process (untargeted metabolomics); however, the missing piece is discussion of molecular testing earlier in this diagnostic process (discussed in more detail below). Considering current healthcare, one aspect that is not addressed is the decreasing number of clinically available tests. The need for tests to be clinically available and not research only is critical for patients; however, because of the rarity of many conditions, these tests are often research only and many tests that were once available clinically are no longer --because labs cannot afford to keep the tests active in their test menus. As testing becomes focused on molecular findings first, this will be an added concern because of inability to determine if variants are pathogenic.
This is a great comment and although beyond the scope of this article we have emphasised this on page 10, lines 368-372. This is a point that is perhaps relevant for future focus in terms of the global network of diagnostic services offering these tests. A current example for our laboratory is that even as an occasional test, Farber enzymology is arguably too rare to offer (though we currently still do).
Other points that are not really mentioned:
Lack of specificity of a specific biomarker to one disease is not critical if molecular testing is done in parallel (or as reflex). We have now emphasised this, page 2, lines 47-48, page 4, lines 149-150, page 10, lines 363-364.
Newborn screening has improved substantially in the last 10 years. The challenge is that the standard testing that is done when NBS is normal (UOA, PAA, ACP) do not provide much diagnostic coverage beyond the current NBS, which covers ~50 conditions. Of course, the NBS varies across countries. We agree and have made comment page 3, line 114 and line 122, page 5, lines 190-192.
As exome first has become the approach to testing--- two outomes/challenges have surfaced: Many VUS are identified in metabolic pathway genes with inability to determine pathogenicity Often, lack of knowledge, training, and experience in metabolism leads to lack of consideration of VUS in the phenotype We agree and have included an extra section (now 4.3) lines 298-308 and an extra figure (Fig. 2) on page 8.
Metabolomics and exome have identified patients with milder metabolic disease who otherwise would have been missed because the phenotype was not classic. (GABA-transaminase deficiency, for example, as shown in Kennedy et al 2019, among others) Thus, the phenotype for metabolic disease is more broad and if limited to what we think we know, we will miss the diagnosis. This has been addressed page 5 lines 305-307.
A few other comments:
UOA methods are not equally sensitive across labs, which may result in missing a diagnosis Acknowledged, page 5, lines 189-192.Line 192: update/add references with ACMG lab guidelines for peroxisomal disorders This has been done, page 6, lines 203-207.
Reference 61 related to Farber-- C26 not so specific to Farber /acid sphingomyelinase deficiency. A reasonable screen but non-specific. Screening for non-specific molecules is still informative—for example, indicating peroxisomal dysfunction-- but not necessarily a specific gene. Narrowing the focus is quite helpful toward diagnosis. Acknowledged and discussed page 10, lines 361-364.
Figure 1 describes a very standard approach to screening that hasn’t changed in decade, without consideration of molecular testing/screening as part of this approach (exome), in addition to impacts of current NBS. Exome first is commonly used. We have emphasised this as a “past” heuristic, and now included the emerging and possible future approaches in this figure and included Figure 2 to address the evolving approach to diagnosing IEM.
X-ALD is now being added to NBS in the U.S. Acknowledged and now referenced in the manuscript.
Reviewer 2 Report
Metabolomics to improve the diagnostic efficiency of inborn errors of metabolism
The authors provide a useful overview of conventional and newer diagnostic metabolomics methods that would be a helpful introduction to the area for a non-expert. I found the writing a little difficult to follow in parts and in some cases potentially misleading. Some general re-editing would be beneficial but some specific rewording is required (listed below). I also feel that some of the drawbacks of non-targeted metabolomics should be mentioned e.g. metabolites of unknown significance (e.g. unknowns that could be treatment artefacts), lack of standardization.
Table 1: other symptoms/systems are affected e.g. renal/nephrocalcinosis. Safer to say “main” symptoms in caption
L112: MMAs are more complex that just congenital B12 deficiency e.g. MUT defects. Please modify
L129 “…. ITS suitability for NBS”
L144 “Specific testing…”?? Do the authors mean “somewhat more specific testing…”? Increased ammonium can still be caused by many IEMs
L169: This sentence is confusing: Do you mean “In the case of acutely presenting conditions, the approach to screening begins with …..”?
L174…: “less apparent patterns” ?? – I think “less specific patterns” is meant. 3MGA’urias are usually quite obvious.
179: “core of the metabolic screen? Practices vary with lab and time. Recommend rewording to: “these techniques have often been the core of a so-called metabolic screen” or similar
180: “Approximately 400” is still far too specific. It’s safer to say many labs report can detect up to hundreds of metabolites
183: Lysosomal enzymology as a first line investigation for DD?!: I don’t think many clinicians would follow this practice nor would labs be able to cope with the work-load. Also, the description of these phenotypes as “specific” is misleading. Please reword this sentence.
199: I don’t think many metabolicians would regard glucose 6-phosphate (not phosphatase) dehydrogenase as a polyol disorder. Please reword
203: PROTEIN glycosylation for clarity
217: it is worth mentioning that the gold standard of enzyme confirmation from a biopsy is often impractical or unethical when very reliable biomarkers or genetic testing offer an alternative confirmation
230: “causes related TO pterin … “
292: “… the number of analytes has not been high-dimensional enough… ” what does this mean? Delete high-dimensional?
351: “These cheminformatic methods are objective and reproducible”. This gives the impression of CVs like core lab analytes. I recommend changing to “…. sufficiently reproducible for screening purposes.”
Author Response
Reviewer 2
The authors provide a useful overview of conventional and newer diagnostic metabolomics methods that would be a helpful introduction to the area for a non-expert. I found the writing a little difficult to follow in parts and in some cases potentially misleading. Some general re-editing would be beneficial but some specific rewording is required (listed below). I also feel that some of the drawbacks of non-targeted metabolomics should be mentioned e.g. metabolites of unknown significance (e.g. unknowns that could be treatment artefacts), lack of standardization.
Table 1: other symptoms/systems are affected e.g. renal/nephrocalcinosis. Safer to say “main” symptoms in caption
This has been amended.
L112: MMAs are more complex that just congenital B12 deficiency e.g. MUT defects. Please modify
Accepted and amended to make more clear, page 3, line 114. Partly the intent of this statement was to allude to the important of this perinatal nutrition-related finding, which is sometimes forgotten in discussions about the impact of MS/MS NBS.
L129 “…. ITS suitability for NBS”
L144 “Specific testing…”?? Do the authors mean “somewhat more specific testing…”? Increased ammonium can still be caused by many IEMs
L169: This sentence is confusing: Do you mean “In the case of acutely presenting conditions, the approach to screening begins with …..”?
Amended, page 5, line 175. There was a degree of conflation between clinical approaches.
L174…: “less apparent patterns” ?? – I think “less specific patterns” is meant. 3MGA’urias are usually quite obvious.
Have amended, page 5, line 181. Although we have had patients identified in our lab with 3MGA-uria that other laboratories missed.
179: “core of the metabolic screen? Practices vary with lab and time. Recommend rewording to: “these techniques have often been the core of a so-called metabolic screen” or similar
Amended page 5, line 186.180: “Approximately 400” is still far too specific. It’s safer to say many labs report can detect up to hundreds of metabolites
Agreed and amended to hundreds, page 5, line 188.
183: Lysosomal enzymology as a first line investigation for DD?!: I don’t think many clinicians would follow this practice nor would labs be able to cope with the work-load. Also, the description of these phenotypes as “specific” is misleading. Please reword this sentence.
Agreed that there is nuance and semantics that are difficult to convey here. By first-tier we mean a first-tier biochemical test rather than a clinical first line. We’ve amended page 6, lines 193-194, to make it more clear, but as mentioned elsewhere, there is considerable variability in clinical approaches due to factors such as funding, proximity to both experts and specialised testing, an issue that much like NGS, metabolomics could help with, if approaches were better standardised.
199: I don’t think many metabolicians would regard glucose 6-phosphate (not phosphatase) dehydrogenase as a polyol disorder. Please reword.
This has been reworded page 6, lines 213-216.
203: PROTEIN glycosylation for clarity
217: it is worth mentioning that the gold standard of enzyme confirmation from a biopsy is often impractical or unethical when very reliable biomarkers or genetic testing offer an alternative confirmation
Agreed and amended page 6, lines 233-235. It’s an important part of the variation in approach to patients that often isn’t captured.
230: “causes related TO pterin … “
292: “… the number of analytes has not been highdimensional enough… ” what does this mean? Delete high-dimensional?
This has been deleted. However, for clarity what was meant was that the number of analytes that are being analysed (high-dimensionality in the sense of a data frame or vector, the basis for many cheminformatics analyses within R and python). An interpretation based on in, say, VLCFAs/BCFAs alone is small enough that classifying as affected/unaffected is fairly simple. If we multiplex these metabolites in a single assay, with say, sphingolipids and organic acids, suddenly even the most capable biochemical geneticists become overwhelmed, particularly as individual analytes may be outside reference intervals.
351: “These cheminformatic methods are objective and reproducible”. This givs the imprifession of ? like core lab analytes. I recommend changing to “…sufficiently reproducible for screening purposes.”
Done.Reviewer 3 Report
The manuscript presents an interesting review of the implementation of mass spectrometry technologies coupled with the expanding field of metabolomics in the diagnosis of inborn errors of metabolism (IEM).
Information are basic and in some points could be enriched with more examples, although describe the major MS based approaches for the determination of metabolic pathway alterations in IEM.
After these general comments, there are only some minor issues that could be addressed:
Minor comments:
In line 123, a bracket in reference 13 needs to be removed.
In line 139, the subsection “Example case: Acute metabolic presentation” could be introduced directly in the text, without the necessity to generate a specific subsection.
In line 305 and 320, the two subsections could be merged in one, or if authors prefer to present in a separate part, it would be appreciating to introduce more than one example for each subsection.
In line 330, the subsection 6.3 are poorly described, and a little more information referred to the cheminformatics and machine learning methods would be necessary.
Resolution in Figure 1 needs to be improved
Author Response
Reviewer 3
The manuscript presents an interesting review of the implementation of mass spectrometry technologies coupled with the expanding field of metabolomics in the diagnosis of inborn errors of metabolism (IEM).
Information are basic and in some points could be enriched with more examples, although describe the major MS based approaches for the determination of metabolic pathway alterations in IEM.
These have been addressed through simplifying/merging sections and through the other reviewers’ specific comments.
After these general comments, there are only some minor issues that could be addressed:
Minor comments:
In line 123, a bracket in reference 13 needs to be removed.
In line 139, the subsection “Example case: Acute metabolic presentation” could be introduced directly in the text, without the necessity to generate a specific subsection.
In line 305 and 320, the two subsections could be merged in one, or if authors prefer to present in a separate part, it would be appreciating to introduce more than one example for each subsection.
Sections have been merged.
In line 330, the subsection 6.3 are poorly described, and a little more information referred to the cheminformatics and machine learning methods would be necessary.
Have amended to elaborate on this page 10, lines 391-392 and page 11, lines 393-399. Combining the clinical and laboratory aspects of IEM has made this a very cross-disciplinary paper, so we kept the analytical component to a minimum initially. The additional comments allude to specific methods, i.e. t-SNE for dimensionality reduction (which we acknowledge hasn’t been universally adopted yet, despite its advantages) and using SVM. There’s lots of potentially interesting discussion about, for instance, using boosted/random forest, but feel it’s outside the scope of this manuscript.
Resolution in Figure 1 needs to be improved.
Lifted from 100 DPI to 600 DPI.